# A neuronal network of mitochondrial dynamics regulates metastasis

M. Cecilia Caino[1,2], Jae Ho Seo[1,2], Angeline Aguinaldo[3], Eric Wait[3], Kelly G. Bryant[1,2], Andrew V. Kossenkov[4], James E. Hayden[5], Valentina Vaira[6,7,8], Annamaria Morotti[7,8], Stefano Ferrero[7,9], Silvano Bosari[7,8], Dmitry I. Gabrilovich[1,10], Lucia R. Languino[1,11], Andrew R. Cohen[3] & Dario C. Altieri[1,2]

The role of mitochondria in cancer is controversial. Using a genome-wide shRNA screen, we now show that tumours reprogram a network of mitochondrial dynamics operative in neurons, including syntaphilin (SNPH), kinesin KIF5B and GTPase Miro1/2 to localize mitochondria to the cortical cytoskeleton and power the membrane machinery of cell movements. When expressed in tumours, SNPH inhibits the speed and distance travelled by individual mitochondria, suppresses organelle dynamics, and blocks chemotaxis and metastasis, *in vivo*. Tumour progression in humans is associated with downregulation or loss of SNPH, which correlates with shortened patient survival, increased mitochondrial trafficking to the cortical cytoskeleton, greater membrane dynamics and heightened cell invasion. Therefore, a SNPH network regulates metastatic competence and may provide a therapeutic target in cancer.

[1] Prostate Cancer Discovery and Development Program, The Wistar Institute, Philadelphia, Pennsylvania 19104, USA. [2] Tumor Microenvironment and Metastasis Program, The Wistar Institute, Philadelphia, Pennsylvania 19104, USA. [3] Department of Electrical and Computer Engineering, Drexel University College of Engineering, Philadelphia, Pennsylvania 19104, USA. [4] Center for Systems and Computational Biology, The Wistar Institute, Philadelphia, Pennsylvania 19104, USA. [5] Imaging Shared Resource, The Wistar Institute Cancer Center, Philadelphia, Pennsylvania 19104, USA. [6] Istituto Nazionale Genetica Molecolare 'Romeo and Enrica Invernizzi', Milan 20122, Italy. [7] Division of Pathology, Fondazione IRCCS Ca' Granda Ospedale Maggiore Policlinico, Milan 20122, Italy. [8] Department of Pathophysiology and Transplantation, University of Milan, Milan 20122, Italy. [9] Department of Biomedical, Surgical and Dental Sciences, University of Milan, Milan 20122, Italy. [10] Translational Tumor Immunology Program, The Wistar Institute, Philadelphia, Pennsylvania 19104, USA. [11] Department of Cancer Biology and Kimmel Cancer Center, Thomas Jefferson University, Philadelphia, Pennsylvania 19107, USA. Correspondence and requests for materials should be addressed to D.C.A. (email: daltieri@wistar.org).

One of the hallmarks of cancer[1] is the rewiring of metabolism from oxidative phosphorylation in mitochondria to aerobic glycolysis, the so-called 'Warburg effect'[2]. This process has been linked to increased metastatic propensity[3], and worse disease outcome, at least in certain tumours[4]. However, mitochondria continue to play an important role in tumour metabolism[5], as oxidative phosphorylation contributes to malignant growth, *in vivo*[6], and fuels unique tumour traits, including repopulation after oncogene ablation[7], and metastatic dissemination[8,9]. How these processes are regulated is unclear, and druggable therapeutic targets of mitochondrial bioenergetics have remained elusive[5].

Recent studies have suggested that mitochondrial metabolism may couple to mechanisms of organelle dynamics, controlling the shape, size and topography of mitochondria in tumour cells[10]. This may be important to sustain highly energy-intensive processes, including the ability of tumour cells to invade across basement membranes[11], the antecedent of metastasis[12]. In this context, the repositioning of energetically active mitochondria to the cortical cytoskeleton of tumour cells[13] has been proposed as a 'spatiotemporal' bioenergetics source to fuel membrane-actin dynamics, turnover of focal adhesion (FA) complexes, and increased chemotaxis and invasion[14].

In this study, we used a genome-wide short hairpin RNA (shRNA) screen to identify novel regulators of mitochondrial function implicated in tumour cell invasion. We found that tumours exploit a cytoskeletal trafficking network well known in neurons to concentrate mitochondria at the peripheral cytoskeleton, and fuel the machinery of cell motility and invasion. A critical regulator of this network is syntaphilin (SNPH), which couples the speed of mitochondrial repositioning with cycles of organelle fusion and fission to regulate metastatic competence in mice, and disease outcome in humans.

## Results

**Novel mitochondrial regulators of tumour cell invasion**. Our previous results demonstrated that inhibition of mitochondrial oxidative phosphorylation with Gamitrinib, a mitochondrial-targeted inhibitor of Heat Shock Protein-90 (Hsp90)[15] abolished tumour cell invasion[8]. To further characterize this pathway, we carried out a genome-wide shRNA screen for molecules that rescued Gamitrinib inhibition of tumour cell invasion (Fig. 1a). Using a cutoff of 500 reads for three independent shRNAs in two separate experiments (Supplementary Fig. 1a), we identified 174 genes that restored tumour cell invasion in the presence of Gamitrinib (Supplementary Table 1). Bioinformatics analysis grouped these molecules as regulators of cytoskeleton, cell adhesion, protein kinases, WD repeat proteins and cell motility (Fig. 1b,c, Supplementary Table 1).

One of the top hits in the screen was SNPH (Fig. 1b), a molecule that arrests mitochondrial trafficking at sites of high-energy demands in neurons[16]. Although previously thought of as 'neuronal-specific'[16], SNPH mRNA was expressed in non-neuronal tissues (Supplementary Fig. 1b), as well as tumour and non-transformed cell types (Supplementary Fig. 1c). To test the relevance of this pathway, we next silenced SNPH mRNA (Supplementary Fig. 1d) and protein (Supplementary Fig. 1e) using multiple, independent small interfering RNA (siRNA) or shRNA sequences. Depletion of SNPH constitutively increased tumour cell invasion (Fig. 1d,e), and reversed the inhibition of tumour cell invasion mediated by Gamitrinib (Supplementary Fig. 1f,g), validating the shRNA screen. In control experiments, cell proliferation over the course of the invasion was unaffected (shRNA) or modestly reduced (siRNA) by SNPH silencing (Supplementary Fig. 1h), and total mitochondria mass per cell

was not affected (Supplementary Fig. 1i). In complementary experiments, stable or transient transfection of tumour cells with SNPH cDNA (Supplementary Fig. 1j) suppressed tumour cell invasion (Fig. 1e).

**SNPH regulation of membrane dynamics of cell motility**. We next looked at the requirements of SNPH regulation of tumour cell motility. In these experiments, depletion of SNPH by siRNA (Fig. 1f, Supplementary Fig. 2a) or shRNA (Supplementary Fig. 2b) increased FA complex dynamics (Fig. 1f, Supplementary Fig. 2a,b), a prerequisite of cell movements[11]. This was associated with decreased number of stable FA complexes, and increased formation of new and decayed complexes (Fig. 1g, Supplementary Fig. 2c). As analysed in two independent tumour cell types, SNPH depletion resulted in overall increased number of dynamic FA complexes (Supplementary Fig. 2d), whereas the total number of FA complexes per cell was not affected (Supplementary Fig. 2e). Consistent with increased FA complex dynamics, silencing of SNPH potently stimulated tumour chemotaxis (Fig. 1h, Supplementary Fig. 3a). This response was associated with faster speed of cell migration (Supplementary Fig. 3b), and greater distance travelled per cell (Supplementary Fig. 3c), without affecting the directionality of cell movements (Supplementary Fig. 3d). Reconstitution of SNPH-depleted cells with siRNA-insensitive SNPH cDNA restored SNPH levels (Supplementary Fig. 3e) and reversed the increase in tumour chemotaxis (Supplementary Fig. 3f), with suppression of both speed of cell migration (Fig. 1i), and distance travelled per cell (Supplementary Fig. 3g).

**SNPH regulation of mitochondrial dynamics**. Mitochondrial trafficking to the cortical cytoskeleton has been proposed as a 'spatiotemporal' energy source to power tumour cell motility and invasion[8]. Consistent with this model, depletion of SNPH stimulated the repositioning of mitochondria from their perinuclear localization to the cortical cytoskeleton of tumour cells (Fig. 2a, Supplementary Fig. 4a). Comparable results were obtained using mitochondria labelled with MitoTracker (Fig. 2b) or Tom20 (Fig. 2c). Reconstitution of SNPH-depleted cells with siRNA-insensitive SNPH cDNA prevented the increased accumulation of mitochondria to the cortical cytoskeleton (Fig. 2d, Supplementary Fig. 4b). Mitochondria in SNPH-depleted tumour cells exhibited faster speed of movements (Fig. 2e,f), higher processivity (time that mitochondria spend in motion/time of analysis; Supplementary Fig. 4c) and greater distance travelled (Fig. 2g,h, Supplementary Movie 1), compared with control transfectants.

In addition to exaggerated mitochondrial trafficking, loss of SNPH was associated with increased mitochondrial dynamics in tumour cells[10] (Supplementary Fig. 5a, Supplementary Movie 2), resulting in higher rates of both fusion and fission events (Fig. 3a, Supplementary Fig. 5b–d). This influenced the speed of subcellular mitochondrial trafficking, as larger mitochondria travelled at $< 0.05\,\mu m\,s^{-1}$, whereas smaller mitochondria moved at up to $0.25\,\mu m\,s^{-1}$ (Fig. 3b). Consistent with these findings, SNPH-depleted cells expressed higher levels of mitofusins (MFN1, MFN2, mitochondrial fusion) (Supplementary Fig. 5e), and increased recruitment of dynein-related protein 1 (Drp1, mitochondrial fission) to mitochondria (Supplementary Fig. 5f), compared with control. Mitochondrial inner membrane fusion effector Opa1 was not affected (Supplementary Fig. 5e), and Drp1 levels in the cytosol were unchanged (Supplementary Fig. 5f). To test the functional relevance of these findings, we next silenced MFN1 or MFN2 by siRNA (Supplementary Fig. 5g) in cells depleted of SNPH. In

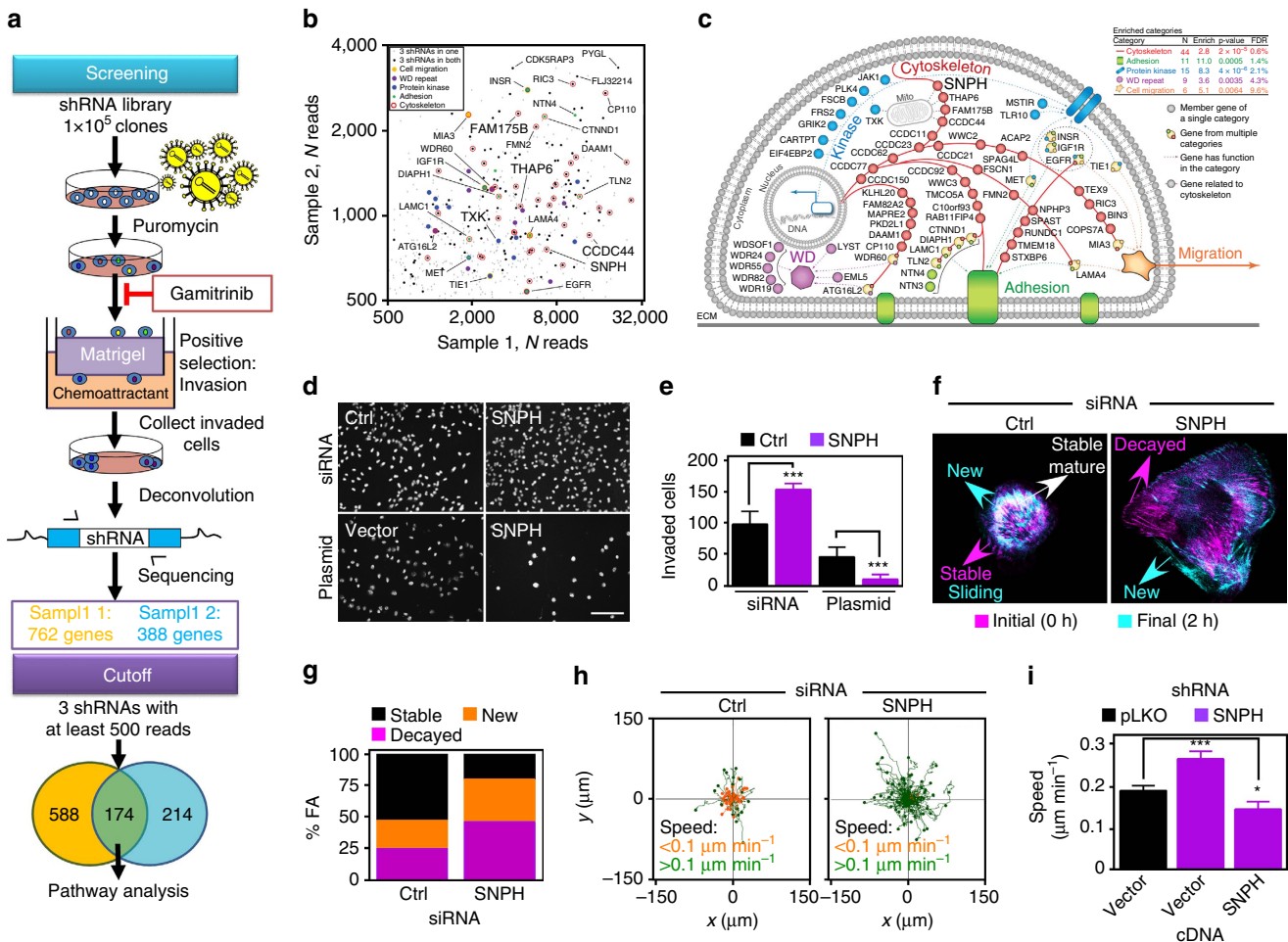

**Figure 1 | SNPH regulation of tumour cell motility.** (**a**) Schematic diagram of the genome-wide shRNA screening to identify novel mitochondrial regulators of tumour cell invasion modulated by Gamitrinib. (**b,c**) Bioinformatics analysis of mitochondrial regulators of tumour cell invasion modulated by Gamitrinib. (**d,e**) PC3 cells transfected with control siRNA (Ctrl) or SNPH-directed siRNA (top) or vector or SNPH cDNA (bottom) were analysed for invasion across Matrigel-coated inserts. Representative micrographs of DAPI-stained cells (**d**) and quantification of invaded cells (**e**). Data are represented as mean ± s.e.m. (n = 3). ***P = 0.0006– < 0.0001 by Student's t test. Scale bar, 200 μm. (**f**) LN229 cells labelled with Talin-RFP and transfected with control (Ctrl) or SNPH-directed siRNA were analysed for FA complex dynamics by time lapse microscopy. Representative merged frames at 0 h (magenta) and 2 h (cyan) are shown. Scale bar, 5 μm. (**g**) FA complex dynamics in LN229 cells transfected with control siRNA or SNPH-directed siRNA (n = 380). (**h**) LN229 cells were transfected with the indicated siRNAs and analysed for 2D chemotaxis by time-lapse video microscopy. Colour-identified cutoff velocities of cell motility are indicated. Each tracing corresponds to an individual cell. Sample sizes were Ctrl (n = 60), SNPH (n = 89). (**i**) shRNA-transduced PC3 cells were reconstituted with vector or SNPH cDNA and analysed for cell motility in a 2D chemotaxis chamber with quantification of speed of cell migration. Data are represented as mean ± s.e.m. Sample sizes were: pLKO,Vector (n = 46), SNPHsh,Vector (n = 48), SNPHsh,SNPHcDNA (n = 46). *P = 0.0209; ***P = 0.0007 by one-way ANOVA and Bonferroni's post-test.

these experiments, depletion of either MFN proteins abolished the increase in tumour cell invasion mediated by SNPH knockdown (Fig. 3c, Supplementary Fig. 5h). Functionally, expression of the antioxidant Peroxiredoxin-III (Prx-III) prevented the increase in MFN1 levels induced by loss of SNPH, whereas mitochondrial superoxide dismutase (SOD2) was ineffective (Fig. 3d,e), and total mRNA levels of MFN1 or MFN2 were unchanged in control or SNPH-depleted cells (Supplementary Fig. 5i).

**A SNPH network regulates tumour cell invasion.** In neurons, SNPH works in concert with a network of molecules that control mitochondrial trafficking (Fig. 3f)[17], and a role of this network in tumour cell movements was next investigated. siRNA silencing of the anterograde kinesin KIF5B (also known as UKHC) (Supplementary Fig. 6a) or atypical mitochondria-associated

Rho GTPase Miro 1 (Supplementary Fig. 6b, left) suppressed the increased tumour cell invasion induced by loss of SNPH (Fig. 3g, Supplementary Fig. 6c). In contrast, silencing of Miro2 (Supplementary Fig. 6b, right) did not reduce tumour cell invasion in SNPH-knockdown cells (Fig. 3g, Supplementary Fig. 6c). Next, we expressed various deletion mutants of SNPH that lack the microtubule binding domain (ΔMBD), kinesin-binding domain (ΔKBD), or LC8-binding domain (ΔLBD) (Supplementary Fig. 6d), and looked at their effects on tumour cell invasion. While expression of full length SNPH suppressed tumour cell invasion (Fig. 3h), ΔMBD, ΔKB, or ΔLBD mutants had no effect (Fig. 3h). Finally, we looked at the role of the SNPH network on mitochondrial trafficking induced by cellular stress, including exposure of tumour cells to inhibitors of the phosphatidylinositol-3 kinase (PI3K) pathway[14]. In these experiments, siRNA silencing of KIF5B, Miro1 or Miro2 all potently suppressed mitochondrial trafficking to the cortical

cytoskeleton (Fig. 3i, Supplementary Fig. 6e), as well as tumour cell invasion (Supplementary Fig. 6f) induced by treatment with the small molecule PI3K inhibitor, PX-866 (ref. 14).

In previous studies, mitochondrial trafficking required energetically active organelles[14]. Here, loss of SNPH was sufficient to deregulate this process, and resulted in robust mitochondrial

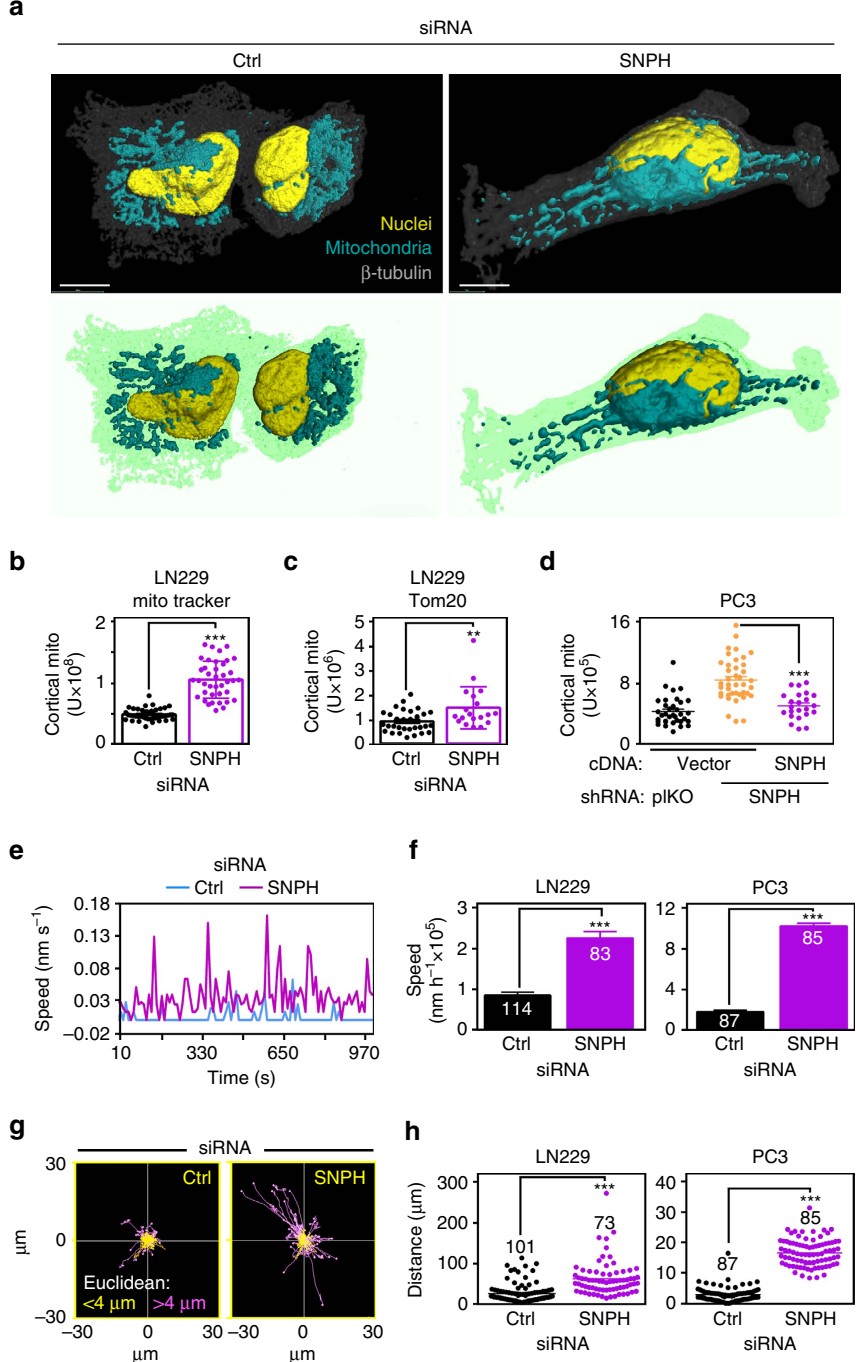

**Figure 2 | Mitochondrial trafficking is modulated by SNPH.** (**a**) LN229 cells transfected with control siRNA (Ctrl) or SNPH-directed siRNA were stained with antibodies against Tom20, β-tubulin and DAPI and analysed by confocal microscopy. Top, 3D rendering of mitochondria localization by fluorescence microscopy. Bottom, 3D drawings represent the cells above, and show mitochondrial repositioning to the cortical cytoskeleton. Scale bar, 10 μm. (**b,c**) Cells were labelled with MitoTracker (**b**) or Tom20 (**c**) and analysed for mitochondrial recruitment to the cortical cytoskeleton. Symbols correspond to individual cells. Data are represented as mean ± s.e.m. **$P = 0.003$; ***$P < 0.0001$ by Student's $t$ test. (**d**) PC3 cells transduced with the indicated shRNA were transfected with vector or shRNA-insensitive SNPH cDNA and analysed for mitochondrial trafficking to the cortical cytoskeleton. Symbols correspond to individual cells. ***$P < 0.0001$ by one-way ANOVA and Bonferroni's post test. (**e,f**) Cells transfected with control or SNPH-directed siRNA and expressing mitochondria-targeted RFP were imaged by time lapse microscopy, and individual mitochondria were tracked through the stack to calculate trafficking parameters (**e**), and the speed of individual mitochondria per condition (**f**). Data are represented as mean ± s.e.m. ($n$ indicated on the individual bars). ***$P < 0.0001$ by Student's $t$ test. (**g,h**) The displacement of mitochondria relative to the initial location was calculated in 2D plots (**g**) and the distance travelled by individual mitochondria was quantified (**h**). Each tracing on **g** represents individual mitochondria and are colour coded according to Euclidean distance displacement. Each dot on **h** correspond to individual mitochondria ($n$ indicated in the individual panels). ***$P < 0.0001$ by Student's $t$ test.

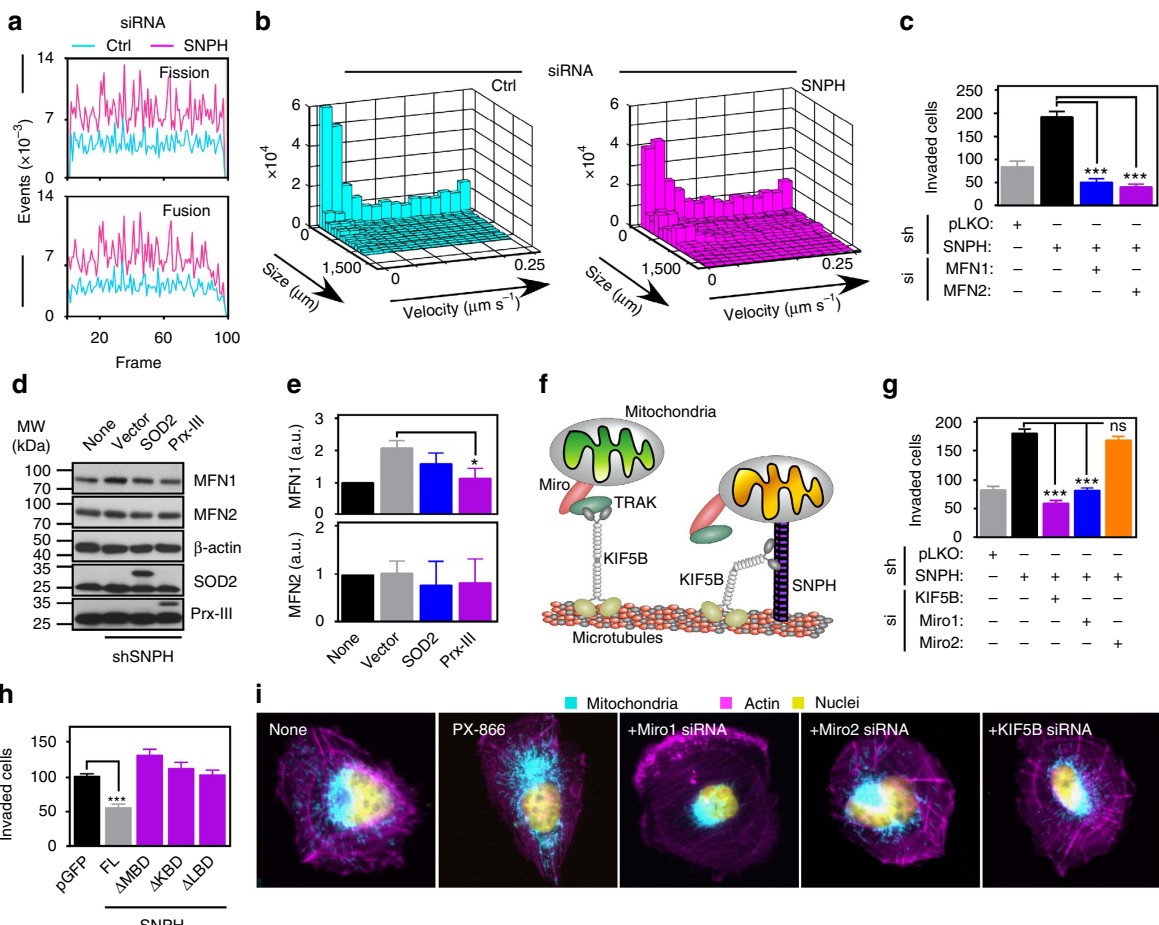

**Figure 3 | SNPH regulation of mitochondrial dynamics.** (**a**) LN229 cells transfected with control siRNA (Ctrl) or SNPH-directed siRNA were labelled with Mito-RFP and mitochondrial fission and fusion events were quantified in individual frames of time lapse microscopy. (**b**) Combined histograms of mitochondrial size and velocity data. Large mitochondria are slow moving (speed $< 0.05\,\mu m\,s^{-1}$) compared with smaller mitochondria (speed up to $0.25\,\mu m\,s^{-1}$). (**c**) PC3 cells stably expressing pLKO or SNPH-targeting shRNA #0 were transfected with siRNA against MFN1 or MFN2 and analysed for Matrigel invasion. Data are expressed as mean ± s.e.m. ($n = 3$). ***$P < 0.001$ by one-way ANOVA and Bonferroni's post test. (**d,e**) PC3 cells stably expressing pLKO or SNPH-targeting shRNA were transfected with cDNAs for SOD2 or PRX-III and analysed by Western blotting (**d**), with quantification of bands by densitometry and normalization to β-actin levels (**e**). Data are expressed as mean ± s.e.m. ($n = 3$).*$P < 0.05$ by one-way ANOVA and Bonferroni's post test. (**f**) Schematic model for mitochondrial trafficking and immobilization in neurons. The implicated molecules in the SNPH network are indicated. (**g**) PC3 cells stably expressing pLKO or SNPH-directed shRNA were transfected with siRNA against KIF5B, Miro1 or Miro2, and analysed for Matrigel invasion. Data are expressed as mean ± s.e.m. ($n = 3$). ***$P < 0.001$ by one-way ANOVA and Bonferroni's post test. ns, not significant. (**h**) PC3 cells were transfected with pGFP, full length (FL) SNPH or SNPH deletion mutants lacking the microtubule-binding domain (ΔMBD), kinesin-binding domain (ΔKBD) or LC8-binding domain (ΔLBD) and analysed for Matrigel invasion. Data are represented as mean ± s.e.m. ($n = 3$). ***$P < 0.0001$ by Student's $t$ test. (**i**) LN229 cells transfected with the indicated siRNAs were treated with the small molecule PI3K inhibitor, PX-866 (5 μM, 48 h) to induce mitochondrial repositioning to the cortical cytoskeleton, and analysed by fluorescence microscopy. Magnification, ×60. None, untreated.

recruitment to the cortical cytoskeleton in nutrient-starved tumour cells (Supplementary Fig. 6g). Similar results were obtained with oxidative phosphorylation-deficient LN229 ρ0 cells, where SNPH depletion induced repositioning of mitochondria to the cortical cytoskeleton, indistinguishably from wild type LN229 cells (Supplementary Fig. 6h,i)[14]. However, mitochondria from LN229 ρ0 cells depleted of SNPH failed to support tumour chemotaxis, with negligible changes in the speed of cell migration (Supplementary Fig. 6j), and overall distance travelled by individual cells (Supplementary Fig. 6k).

**SNPH is a novel metastasis suppressor**. To examine the impact of SNPH on tumour progression, we next inspected clinically annotated public databases. We found that SNPH was mostly downregulated in epithelial and haematologic malignancies, compared with normal tissues (Fig. 4a). In contrast, Miro2 levels,

which support mitochondrial trafficking (Fig. 3i, Supplementary Fig. 6e) and tumour cell invasion (Supplementary Fig. 6f), were typically upregulated in the same tumours (Fig. 4a). When examined for disease outcome, loss of SNPH was an unfavourable prognostic factor, correlating with shortened overall survival in adenocarcinoma of the colon (Supplementary Fig. 7a) and lung (Supplementary Fig. 7b).

To independently validate these results, we examined primary patient samples in a universal cancer tissue microarray, by immunohistochemistry. Consistent with the data above, SNPH was expressed in normal breast epithelium, but downregulated or undetectable in breast cancer (Fig. 4b,c), as well as adenocarcinoma of the prostate (Supplementary Fig. 7c), colon and lung (Supplementary Fig. 7d). Glioblastomas expressed high levels of SNPH, compared with normal brain (Supplementary Fig. 7d). When analysed in a series of clinically annotated breast cancer patients ($n = 324$, Supplementary Table 2), SNPH levels

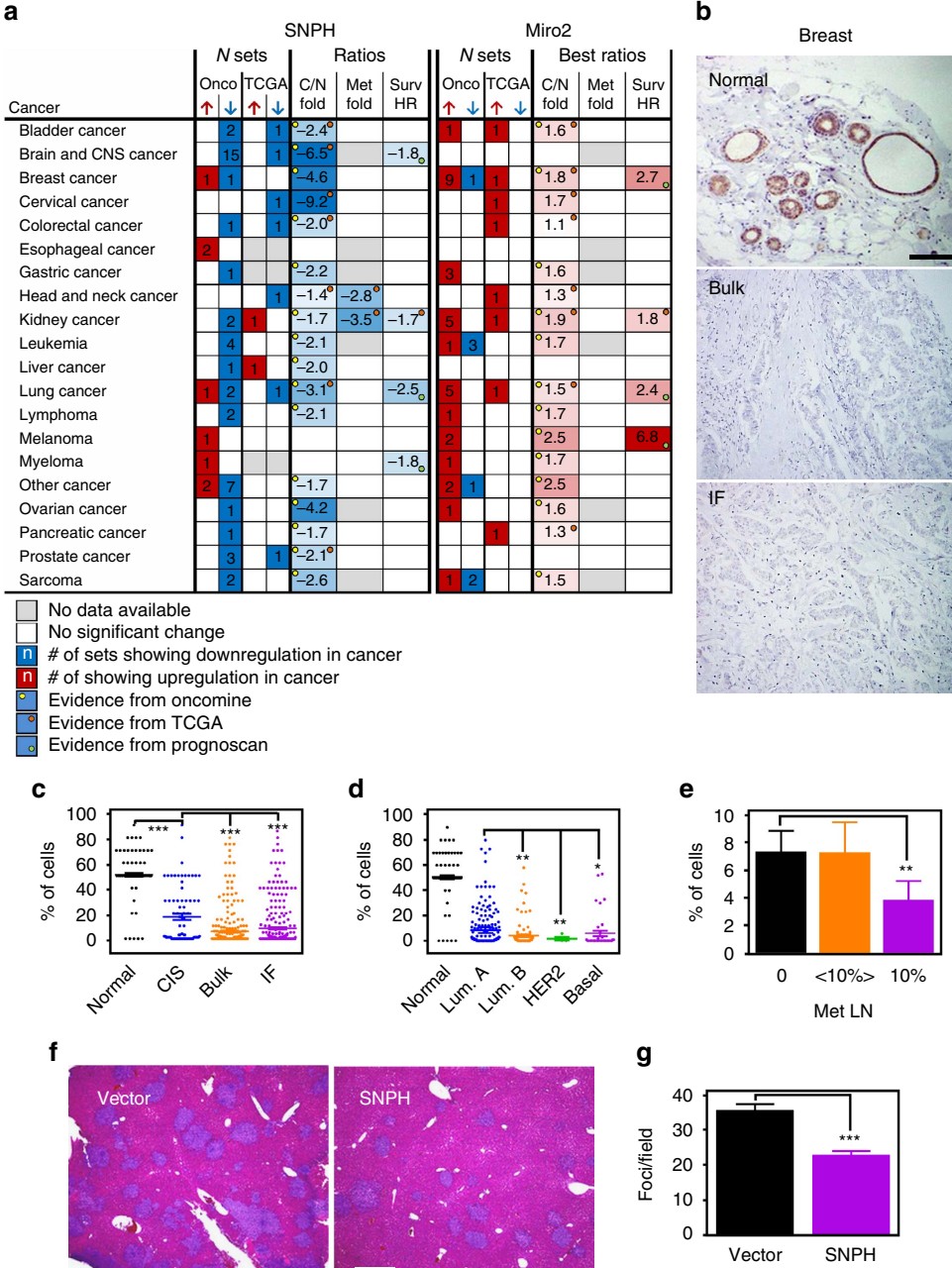

**Figure 4 | SNPH controls tumour progression and metastasis.** (**a**) Bioinformatics analysis of SNPH or Miro2 differential expression in cancer versus normal tissues in public databases (Oncomine, TCGA, Prognoscan). ↑, upregulation; ↓, downregulation; C/N, cancer/normal; HR, hazard ratio. (**b**) IHC analysis of SNPH expression in breast carcinoma versus normal breast epithelium. Bulk, main lesion; IF, invasive front. Scale bar, 100 μm. (**c**) SNPH-positive cells in normal breast, CIS, bulk breast carcinoma or invasive front (IF). Each point corresponds to individual patients. \*\*\*$P < 0.0001$ by unpaired $t$ test with Welch's correction. (**d**) SNPH-positive cells in molecular subtypes of breast cancer. Lum, luminal. HER2, HER2-enriched. Each point corresponds to individual patients. \*\*$P = 0.006$–$0.007$; \*$P = 0.01$ by unpaired $t$ test with Welch's correction. (**e**) SNPH-positive cells in breast cancer with no lymph node metastasis, $\leqslant 10\%$ of metastatic lymph nodes ($<10\%$) or $>10\%$ metastatic lymph nodes ($>10\%$). Data are represented as mean ± s.e.m. \*\*$P = 0.007$ by Mann Whitney test. (**f**) Histologic analysis of liver metastases after intrasplenic injection of PC3 cells stably expressing vector or SNPH cDNA. Scale bar, 500 μm. (**g**) Number of liver metastatic foci per field in animals from **f**. Data are represented as mean ± s.e.m. ($n = 15$). \*\*\*$P < 0.0001$ by Student's $t$ test.

were more prominently downregulated in the invasive front and bulk (core) lesions, compared with ductal carcinoma *in situ* (Fig. 4c). Histologic subtypes of HER2-enriched and basal-like carcinoma also contained lower levels of SNPH than luminal A or luminal B cases (Fig. 4d). SNPH expression in breast cancer was independent of stage (Supplementary Fig. 7e) or histotype (Supplementary Fig. 7f), but was significantly more downregulated in metastatic tumours to loco-regional lymph

nodes, compared with non-metastatic or low-metastatic cases (Fig. 4e).

Based on these data, we next asked if SNPH influenced metastasis in a mouse model of disseminated disease, *in vivo*[8]. In these experiments, intrasplenic injection of PC3 cells stably transfected with vector gave rise to extensive metastatic dissemination to the liver 11 days after animal reconstitution (Fig. 4f,g). In contrast, stable overexpression of SNPH in these

cells reduced the number (Fig. 4f,g), and size (Supplementary Fig. 7g) of metastatic foci to the liver, lowering the overall metastatic burden of reconstituted mice (Supplementary Fig. 7h).

## Discussion

In this study, we have shown that tumours reprogram a network of mitochondrial dynamics well characterized in neurons to fuel cell motility, invasion and metastasis. This pathway is centred on molecules that promote (Miro1/2, KIF5B) or suppress (SNPH) the trafficking of mitochondria to the cortical cytoskeleton of tumour cells, a process that provides a concentrated, 'spatiotemporal' energy source to fuel membrane dynamics and cell movements. As a negative regulator of this pathway, SNPH acts as a conceptually novel 'metastasis suppressor', progressively downregulated or lost during tumour progression, and associated with heightened mitochondrial dynamics, increased tumour cell invasion and shortened patient survival.

The role of mitochondrial bioenergetics in cancer has been controversial. The pervasive glycolytic metabolism that is hallmark of most malignancies[2], and the characterization of 'oncometabolites' produced by a dysfunctional or mutant respiratory chain[18] have suggested that mitochondrial metabolism may be dispensable in malignancy, and may actually function as a 'tumour suppressor', at least in certain tumours[19]. It is now clear, however, that oxidative phosphorylation remains an important driver of tumour growth[6], fuelling unique traits especially important for disease progression. There is evidence that one of these traits is tumour cell motility and metastasis[8,9], but the requirements of this response have been elusive, variously linked in previous studies to aberrant ROS production[20], increased fatty acid metabolism[21] or remodelling of oxidative phosphorylation complexes[22].

The data presented here suggest an alternative scenario, where the coupling of mitochondrial bioenergetics to mechanisms of organelle dynamics[10] helps reposition mitochondria to membrane protrusions of cell motility[11], fuelling chemotaxis and increased cell invasion[14]. Although this process has been often considered 'neuronal-specific'[23], we now know that mitochondria travel along the microtubule network also in non-neuronal cells, supporting the motility of normal[24], as well as transformed[13] cell types in response to stress conditions of the microenvironment[14].

Unlike neurons, where SNPH arrests mitochondrial movements along the cytoskeleton[16], tumour-associated SNPH exhibited new functions in this process, reducing the speed and distance travelled by individual mitochondria, and suppressing cycles of mitochondrial fission and fusion, that is, dynamics[25]. Mitochondrial dynamics has recently emerged as a driver of malignancy[10], contributing to oncogene-directed transformation[26], tumour progression[27] and cell invasion[28]. Using a novel algorithm that tracks quantitative changes in mitochondrial morphology, we have shown here that loss of SNPH heightened both fusion and fission events in tumour cells, consistent with modulation of key effectors of organelle dynamics, including increased expression of MFN1 and recruitment of Drp1 to mitochondria[10]. Although this process controlled the velocity of mitochondrial trafficking, it is also possible that SNPH modulation of dynamics couples to organelle quality control mechanisms[29], preventing the repositioning of inactive or damaged mitochondria to the cortical cytoskeleton. Consistent with this model, loss of SNPH was sufficient to allow recruitment of de-energized mitochondria from LN229 ρ0 cells[14] to the cortical cytoskeleton, whereas silencing of MFN1 or MFN2 suppressed mitochondrial trafficking and tumour cell invasion.

Initial structure–function studies presented here demonstrated that specialized SNPH domains implicated in microtubule binding, kinesin association or LC8 interaction[17] were required for mitochondrial trafficking in tumour cells. In addition, molecules known to regulate mitochondrial repositioning in neurons, including KIF5B[30] and Miro1 (ref. 24) emerged here as novel modulators of organelle movements in the SNPH pathway in tumours. Conversely, another atypical mitochondrial GTPase, Miro2 (ref. 23) did not affect SNPH-dependent functions, but was still required to support mitochondrial trafficking induced by cellular stress, that is, exposure to molecular therapy[14]. Together, this suggests that Miro2 contributes to mitochondrial trafficking in transformed cells independently of SNPH, and this may be important in vivo, as Miro2 levels are consistently increased in disparate tumour types from bioinformatics analysis of public databases.

A combination of quantitative cell motility and invasion approaches in vitro, together with a mouse model of liver metastasis that recapitulates key steps of disease dissemination in vivo, established a role of SNPH as a conceptually novel 'metastasis suppressor'. Consistent with this model, analysis of public databases as well as primary breast cancer patients demonstrated that SNPH levels are consistently downregulated or lost during tumour progression, correlating with increased metastasis, and shortened patient survival. The mechanism(s) that titrate the levels of SNPH during tumour progression remains to be elucidated. However, one possibility is that this process hinges on changes in the tumour microenvironment, which may force differential allocation of often limited resources to support either cell proliferation or cell motility, but not both[31]. Specifically, rising hypoxia and chronic nutrient deprivation, as typically seen in advanced tumours[32], may reprogram mitochondrial dynamics through the SNPH pathway to support heightened cell motility and 'escape' from an unfavourable ecosystem[33]. This possibility is consistent with recent data that mitochondrial metabolism is required to support tumour cell motility under conditions of nutrient deprivation[8], and that cellular stress introduced by molecular therapy potently stimulates mitochondrial trafficking to the cortical cytoskeleton and increased cell invasion[14].

In sum, we have shown that tumours reprogram a neuronal network of mitochondrial trafficking and dynamics to control metastatic competence. While these data reinforce a role of mitochondrial metabolism as an important driver of tumour progression[4], molecules in the SNPH pathway exploited in cancer, including KIF5B and Miro1 are druggable, opening fresh therapeutic opportunities to oppose metastatic dissemination in patients.

## Methods

**Antibodies and reagents.** Antibodies to Drp1 (clone D6C7, Cell Signaling#8570 diluted 1:1,000), MFN1 (clone D6E2S, Cell Signaling#14739 diluted 1:1,000), MFN2 (clone D2D10, Cell Signaling#9482 diluted 1:1,000), OPA1 (clone 18BD BD Laboratories#612606 diluted 1:1,000), PrxIII (clone 4G10, Santa Cruz Biotechnology sc-59663 diluted 1:1,000), SOD2 (clone D3X8F, Cell Signaling# 13141 diluted 1:1,000), SNPH (Sigma#HPA049393 diluted 1:1,000), SNPH (H-250, Santa Cruz Biotechnology#sc-33824 diluted 1:500), KIF5B (Millipore#ABT405 diluted 1:1,000), Miro1 (RhoT1 A-8 Santa Cruz#sc-398520 diluted 1:200), Miro2 (RhoT2 H-45 Santa Cruz#sc-135387 diluted 1:500), FLAG (clone M-2, Sigma#F1804 diluted 1:5,000) and β-actin (clone AC-15 Sigma#A5441 diluted 1:100,000) were used for Western blotting. Antibodies to paxillin (clone 5H11 Millipore/Upstate#05-417 diluted 1:100), α-tubulin (clone DM1A Sigma# T9026 diluted 1:200), Tom20 (FL-145Santa Cruz# sc-11415 diluted 1:250), SNPH (Sigma#HPA049393 diluted 1:500) and anti-mitochondria antibody (MTC02 Abcam# ab3298 diluted 1:500) were used for immunofluorescence. The small molecule pan-PI3K inhibitor PX-866 was from LC Laboratories. The complete chemical synthesis, HPLC profile, and mass spectrometry of mitochondrial-targeted Hsp90 small molecule antagonist and suppressor of oxidative phosphorylation[34], Gamitrinib has been reported[15]. The Gamitrinib variant containing triphenylphosphonium as a mitochondrial-targeting moiety[15] was used for the shRNA screen at non-toxic concentrations. Mitotracker dyes, Phalloidin Alexa Fluor 488, CellLight Talin-RFP BacMam 2.0, CellLight Mito-RFP BacMam

2.0 and secondary antibodies for immunofluorescence were from Molecular Probes.

**Cell culture.** Human glioblastoma LN229, prostate epithelial (RWPE-1), prostate adenocarcinoma (LNCaP, C4-2, DU145, PC3), breast epithelial (MCF10A), breast adenocarcinoma (MDA-MB-231), human diploid fibroblasts MRC-5 and mouse fibroblasts NIH3T3 were obtained from the American Type Culture Collection (ATCC, Manassas, VA), and maintained in culture according to the supplier's specifications. Benign prostate hyperplasia (BPH1) cells were a generous gift from Dr Simon W. Hayward (Vanderbilt University). A highly metastatic clone of PC3 with bone tropism (PC3-ML) was a generous gift from Dr Alessandro Fatatis (Drexel University College of Medicine). Cell passaging was limited to <40 passages from receipt and cell lines were authenticated by STR profiling with AmpFlSTR Identifiler PCR Amplification Kit (Life Technologies) at the Wistar Institute's Genomics facility. Mycoplasma free-cultures were confirmed at the beginning of the studies, and every 2 months afterwards, by direct polymerase chain reaction (PCR) of cultures using Bioo Scientific Mycoplasma Primer Sets (cat#375501) and Hot Start polymerase (QIAGEN). Mitochondrial DNA-deficient LN229 ρ0 cells were described previously[14] and maintained in culture medium supplemented with $80 \, \text{ng ml}^{-1}$ uridine and 1 mM sodium pyruvate. Conditioned media was prepared from exponentially growing cultures of NIH3T3 cells as described four in DMEM supplemented with $4.5 \, \text{g l}^{-1}$ D-glucose, sodium pyruvate, 10 mM HEPES and 10% FBS for 48 h. Where indicated, glucose starvation was carried out for 24 h in DMEM without glucose supplemented with dialyzed FBS. For experiments where cells were cultured oxidative medium (Gal medium), DMEM without glucose was supplemented with 25 mM D-( + )-galactose and dialyzed FBS.

**shRNA genome-wide screening.** A positive selection-based genome-wide screen was carried out in duplicate by transducing PC3 cells with the pLKO.1 based-TRC lentiviral shRNA library[35,36]. The $1 \times 10^7$ clones encoded by the library were divided into 12 pools of ∼1,800 genes each, with an average of 5 clones per gene (9,000 clones per pool). Cells were transduced with one lentiviral particle every 3 cells (m.o.i. = 0.3) at a representation of 1,000 cells per clone = $9 \times 10^6$ cells per pool. Following 4 days of selection in the presence of $2 \, \mu \text{g ml}^{-1}$ puromycin, transduced cells were subject to treatment with Gamitrinib ($5 \, \mu \text{M}$, 16 h) and seeded for invasion assays using Matrigel-coated 8 μm pore size six-well Transwells (BD) at a density of $3.4 \times 10^5$ cells cm$^{-2}$ of membrane. NIH3T3 medium was used as chemoattractant. The following day, invasive cells were recovered by trypsinization of the lower compartment of the transwell, and used for genomic DNA preparation using QIAamp DNA Mini kit (QIAGEN), according to the manufacturer's protocol for cultured cells. Next, shRNAs were amplified from genomic DNAs, cleaved into half-hairpins by digestion with XhoI and batch sequenced by Next Generation Sequencing (NGS, see below).

**Deconvolution and bioinformatics analysis of the shRNA screen.** The shRNAs inserted into the genomic DNA of invasive cells were amplified by PCR using the strategy described[35], except that the reverse primer was altered to have a one base pair (bp) mutation of the XhoI site located in the pLKO.1 vector (extended data Fig. 1a). With this modification, XhoI digestion of the PCR product renders only two fragments of 111 and 70 bp. The sequence of the primers used was Forward: 5′-TGGCTTTATATATCTTGTGGAAAGGACGAAACACCG-3′ and Reverse (XhoI site mutated): 5′-TGTGGATGAATACTGCCATTTGTCACGAGGTC-3′. PCR amplification reactions were carried out starting with 12 μg of genomic DNA from each pool with TaKaRa Ex Taq reagents (Clontech). PCR products (shRNA hairpins) from the 12 pools of each sample were further combined, and digested overnight with XhoI at 37 °C. The 111 bp bands were purified from a 4% MetaPhor agarose (Cambrex) gel. Five hundred ng of purified half-hairpin were used for paired-end Illumina cDNA library preparation (TruSeq v.2). The libraries were run on a Beckman Illumina HiSeq instrument producing 99,983,355 and 98,756,442 quality reads, and further processed for bioinformatics data analysis.

For bioinformatics analysis, samples were sequenced and processed using custom script that inspected only reads with base call quality of at least 35 for pattern [read start]TCGAG[21 bp-variable-insert] 5′-CCGGTGTTTCGTCCTTTC CACAAG-3′. Resulting inserts were mapped to 'TRC shRNA Library Inventory Flatfile' (version from 5 April 2011) and the number of reads for each shRNA was counted. Genes that had at least three shRNAs with at least 500 reads were considered significant in an experiment, and only genes that were significant in both experiments were considered for further analysis. Enrichment analysis of candidate genes was done using DAVID software (https://david.ncifcrf.gov) searching for overrepresentation of: gene ontology molecular functions, biological processes and cellular components as well as swiss-prot functional keywords and KEGG pathways. Results that passed false discovery rate <10% were called significant. Significantly enriched related categories were grouped, and a general group name was assigned and reported along with total number of unique genes combined across all categories from the group, minimal P value and average false discovery rate (Supplementary Table 1).

**Plasmids and transfections.** True-ORF pCMV6-Entry-myc-Flag encoding SNPH (cat# RC207749), SOD2 (cat# RC202330) and PrxIII (cat# RC205080) were from Origene. SNPH mutants lacking the microtubule binding domain (ΔMTB, Δ86-159 aa), LC8-binding domain (ΔLC8, Δ311-317 aa) or kinesin-binding domain (ΔKBD, Δ337-422 aa) were generated with the Stratagene QuikChange II XL Site-Directed Mutagenesis Kit (Agilent Technologies), and confirmed by DNA sequencing. Cells were transfected with 2 μg of pcDNA plus 4 μl X-treme gene HP (Roche) for 24 h in complete medium, washed and subject to the indicated treatments. Cells stably expressing full length SNPH, SNPH mutants or an empty vector were generated by transfection of plasmids, followed by a 2-week selection in the presence of $1 \, \text{mg ml}^{-1}$ G418 (Geneticin).

**Gene silencing.** Gene knockdown experiments by siRNA were carried out as described[8]. The following sequences were used: control, non-targeting siRNA pool (Dharmacon, D-001810) or specific siRNA pools targeting SNPH (either Dharmacon L-020417 or Santa Cruz sc-41369), KIF5B (Santa Cruz, sc-36777), Miro1 (Santa Cruz, sc-93809), Miro2 (Santa Cruz, sc-93029), Mfn1 (Santa Cruz sc-43927) or Mfn2 (Santa Cruz sc-43928). Tumour cells were transfected with the individual siRNA pools at 30–60 nM concentration in the presence of Lipofectamine RNAiMAX (Invitrogen) at a 1:1 ratio (vol siRNA 20 μM:vol Lipofectamine RNAiMAX). After 48 h, transfected cells were validated for target protein knockdown by Western blotting or qPCR, and processed for subsequent experiments. Alternatively, three independent shRNA sequences were used for targeting the 3′UTR of human SNPH: TRCN 0000147900, TRCN 0000128545 and TRCN 0000150197 (Wistar Molecular Screening Facility). An empty pLKO-based lentivirus was used as control. PC3 cells stably expressing shRNA targeting SNPH were generated by infection with lentiviral particles, followed by a 2-week selection in the presence of puromycin at $2 \, \mu \text{g ml}^{-1}$.

**Protein analysis.** For Western blotting, protein lysates were prepared in 20 mM Tris HCl, pH 7.5, 137 mM NaCl, 1% Triton X-100, 10% glycerol containing EDTA-free Protease Inhibitor Cocktail (Sigma) and Phosphatase Inhibitor Cocktail PhosStop (Roche), sonicated and precleared by centrifugation at 14,000g for 10 min at 4 °C. Equal amounts of protein lysates were separated by SDS gel electrophoresis, transferred to polyvinylidene difluoride membranes, blocked in 5% low fat milk diluted in TBST buffer (20 mM Tris HCl, pH 7.5, 150 mM NaCl, 0.1% Tween-20), and further incubated with primary antibodies diluted in 5% BSA/TBST for 18 h at 4 °C. For SNPH (Sigma#HPA049393), membranes were incubated with primary antibody diluted 1:1,000 in 5% low fat milk diluted in TBST for 1 h at 22 °C. For detection of β-actin (clone AC-15 Sigma#A5441) membranes were incubated with primary antibody diluted 1:100,000 in 5% BSA diluted in TBST for 30 min at 22 °C. After washing in TBST, membranes were incubated with HRP-conjugated secondary antibodies (1:1,000–1:5,000 dilution in 5% BSA/TBST) for 1 h at 22 °C, washed with TBST and protein bands were visualized by enhanced chemiluminescence. Full blots from main and Supplementary Figs are shown in Supplementary Fig. 8.

**mRNA analysis.** Relative mRNA levels of SNPH were determined by qPCR. RNA was extracted with PureLink RNA Mini Kit (Life Technologies) following the in-column DNA digestion protocol. Alternatively, RNA from eight normal human tissues was obtained from BioChain and digested with RNAse-free DNAseI (Thermo Scientific). Two μg of RNA were reversed transcribed using random hexamers for 1 h at 37 °C using the TaqMan High-Capacity cDNA Reverse Transcription Kit (Life Technologies#4368814). One μl of cDNA diluted 1:5 was used as template for qPCR reactions with TaqMan Gene Expression (GEX) assays to detect the various transcripts in a ABI7500 Fast Real Time PCR system. The following GEX assays were obtained from TaqMan (Life Technologies): SNPH (Hs00920132_m1), MFN1 (Hs00966851_m1), MFN2 (Hs00208382_m1). ACTB (Hs99999903_m1) and GAPDH (Hs99999905_m1) assays were used as endogenous control to normalize the levels of mRNA across samples. The relative abundance of mRNA was calculated according to the ΔΔCt method.

**Mitochondria time-lapse microscopy.** Cells ($4 \times 10^4$) growing in high optical-quality glass bottom 35-mm plates (MatTek Corporation) were transduced with Mito-RFP BacMam virus (50 particles per cell) for 18 h, then with Tubulin-GFP BacMam virus (Life Technologies) (50 particles per cell) for 18 h, washed and imaged with a × 63 1.40 NA oil objective on a Leica TCS SP8 X inverted laser scanning confocal microscope. Short duration time-lapse sequences were carried out in a Tokai Hit incubation chamber equilibrated to 37 °C and 5% CO$_2$ with bidirectional scanning at 8,000 Hz using a resonant scanner. Time lapse was performed at 10 s per frame for 1,000 s (LN229), or 1 s per frame for 300 s (PC3). Individual 12-bit images were acquired using a white-light supercontinuum laser (2% at 557 nm and 5% at 488 nm) and HyD detectors at × 3 digital zoom with a pixel size of 120 nm × 120 nm. A pinhole setting of 3 Airy Units provided a section thickness of 2.042 microns. At least 15 single cells per condition were collected from three independent experiments.

**Analysis of mitochondria trafficking.** Time-lapse sequences were imported into FIJI and individual mitochondria were manually tracked using the Manual Tracking plugin. Mitochondria were tracked along the stacks until a fusion event prevented continued tracking. The speed and distance between frames were used to calculate the mean speed and cumulative distance travelled by each individual mitochondria. Processivity was defined as the proportion of time mitochondria spent in motion relative to the total time of tracks and calculated based on the frame-to-frame data for speed. Any given frame with speed $\neq 0$ was considered as 'in motion'. Based on processivity, mitochondria were further classified into high processivity ($>70\%$ of time spent in motion) or low processivity ($<70\%$ of time spent in motion).

**Computational detection of fission and fusion events.** Computational image analysis tasks to quantify mitochondrial dynamics included image denoising, organelle segmentation and multi-target tracking. The denoising algorithm used a model-based approach, combining a low-pass filter (diameter of 9 pixel) to remove slow varying background with a median filter (diameter of 5 pixels) to remove high-frequency shot noise. This approach was originally proposed for protein localization[37], and later applied to images of synthetic organelle data[38] and three-dimensional (3D) multi-channel time-lapse images showing stem cell proliferation[39]. The mitochondrial segmentation algorithm combined the detection of intensity peaks on the denoised image with a thresholded foreground region obtained from an adaptive Otsu transform[38,40]. The peaks within the foreground regions identify mitochondrial locations, whereas the connected component of foreground pixels associated with each mitochondrial peak provide the size of the individual or aggregate mitochondria to be tracked.

The applied tracking algorithm solved the data association problem between pairs of frames in a technique based on the Hungarian assignment algorithm, an approach originally developed for tracking endosomes in live-cell imaging[41]. More advanced tracking algorithms that solve the data association problem over multiple image frames simultaneously[40,42] were less effective for this data set due to the frequent fission and fusion events. The single parameter required by the tracking algorithm is maximal mitochondrial velocity, empirically estimated at $0.32 \, \mu m \, s^{-1}$ (25 pixels per frame). Following tracking, tracks of length shorter than an empirically determined threshold of three image frames were removed.

Fission and fusion events were identified from the tracking results by identifying conditions where a mitochondria track either ends (fusion) or appears (fission) in a spatiotemporal region that enables associating that track with another existing track. The number of fission and fusion events in each image frame were normalized to the number of foreground mitochondrial pixels to allow comparison of fission and fusion events among cells of different size and mitochondrial volume. These normalized counts were treated as a sample of a random variable, and compared between SNPH and control populations using non-parametric two sample Kolmogorov–Smirnov tests for differences of distribution, and Wilcoxon rank sum tests for differences of medians.

**Focal Adhesion dynamics.** Analysis of FA dynamics was carried out as described[14]. Cells growing in high optical quality 35-mm glass bottom plates (MatTek Corporation) were transduced with Talin-RFP BacMam virus (50 particles per cell) for 18 h and imaged with a ×40 objective on a Leica SP8 confocal microscope. The atmosphere was equilibrated to 37 °C and 5% $CO_2$ in an incubation chamber. Time-lapse microscopy was carried out for the indicated time intervals at 3 min per frame. Sequences were imported into ImageJ for further analysis. The initial and final frames were duplicated and assembled as composite images. FA complexes were manually counted and classified (according to the presence in some or all of the time frames) into four groups: decaying, newly formed, stable sliding (FA moves to a different position over time) and stable mature (merged areas). The rate of decay and assembly of FA complexes was calculated for each cell as the number of FA changing per hour. At least 200 FA complexes from 10 cells were analysed from 5 to 10 independent time-lapse videomicroscopy experiments per condition.

**Immunofluorescence.** Tumour cells under the various conditions tested were fixed in formalin/PBS (4% final concentration) for 15 min at 22 °C. Cells were then permeabilized with 0.1% Triton X-100/PBS for 5 min, washed, and incubated in 5% normal goat serum (NGS, Vector Labs) diluted in 0.3 M glycine/PBS for 60 min. Primary antibodies against paxillin (diluted 1:100), Tom20 (diluted 1:300), α-tubulin (diluted 1:200) or MTC02 (diluted 1:500) were added in 5% NGS/0.3 M glycine/PBS and incubated for 18 h at 4 °C. After three washes in PBS, secondary antibodies conjugated either to Alexa488, TRITC or Alexa633 diluted 1:500 in 5% NGS/0.3 M glycine/PBS and added to cells for 1 h at 22 °C. Where indicated, F-actin was stained with phalloidin Alexa488 (1:200 dilution) for 30 min at 22 °C. Slides were washed and mounted in 4,6-diamidino-2-phenylindole (DAPI)-containing Prolong Gold mounting medium (Life Techonologies). At least seven random fields were analysed by fluorescence microscopy in a Nikon i80 microscope.

**Cortical mitochondria and total mitochondria mass quantification.** Mitochondria/F-actin composite images were analysed in ImageJ as described[14].

Briefly, the F-actin channel was used to manually label the cell boundary and a belt extending from the boundary towards the inside of the cell was marked as the 'cortical mask'. This cortical mask was subsequently applied to the mitochondria channel to measure intensity at the cortical region, and normalized to total mitochondria intensity per cell and cell area. A minimum of 30 cells were analysed in each independent experiment to obtain mean values. For quantification of total mitochondrial mass, composite images were analysed in ImageJ. The cell border was manually traced on the F-actin channel and this 'cell mask' was subsequently applied to the mitochondria channel to measure total mitochondria signal per cell. Maximum intensity was monitored to ensure no pixel saturation (for example, max intensity $<256$ for 8-bit images). Mitochondria mass was normalized to total cell area. A minimum of 30 cells were analysed in each independent experiment to obtain mean values. As a second independent method to determine mitochondria mass, $2 \times 10^4$ cells plated in quadruplicate on 96-MW plates were labelled 1 h at 37 °C with 50 nM Mitotracker Green FM (Life Technologies), washed and green fluorescence was read in a multiplate reader using a FITC filter. Mitochondria mass (green fluorescence) per well was normalized to total cell number.

**Mitochondrial isolation.** Mitochondrial fractions were prepared using a mitochondrial isolation kit for cultured cells (ThermoFisher). Briefly, cells were homogenized by 70 strokes using a Dounce Grinder in isolation buffer A plus protease inhibitor cocktail. Cell extracts were collected into equal volumes of isolation buffer C with buffer A. Cell debris and nuclei were removed by centrifugation at 700g for 10 min, and mitochondrial fractions were collected by centrifugation at 3,000g for 25 min.

**Two-dimensional tumour cell chemotaxis.** Experiments were carried out as described[14]. Briefly, $1 \times 10^4$ cells were seeded in two-dimensional (2D) chemotaxis chambers (Ibidi) in complete medium. After 6 h attachment, cells were washed and the reservoirs were filled with serum-free medium supplemented with 0.1% BSA, followed by gradient setup using NIH3T3 conditioned medium. Videomicroscopy was performed over 10 h, with a time-lapse interval of 10 min. Stacks were imported into Image J for analysis. Images were aligned according to subpixel intensity registration with the StackReg plugin for Image J[43]. At least 30 cells were tracked using the Manual Tracking plugin for Image J, and the tracking data from four independent time-lapse experiments were pooled and exported into the Chemotaxis and Migration Tool v2.0 (Ibidi) for graphing and calculation of mean and s.d. of speed, accumulated distance and Euclidean distance of movement.

**Tumour cell invasion.** Experiments were carried out essentially as described[8] using Growth Factor Reduced Matrigel-coated 8 μm PET transwell chambers (Corning). Tumour cells were seeded in duplicates onto the coated transwell filters at a density of $0.5–1 \times 10^5$ cells per well in medium containing 0.8% FBS, and conditioned media from NIH3T3 was placed in the lower chamber as chemoattractant. Cells were allowed to invade for 16–24 h, non-invading cells were scraped off the top side of the membranes and the invasive cells on the transwell insert were fixed in methanol. Membranes were mounted in medium containing DAPI (Vector Labs) and analysed by fluorescence microscopy. Five random fields at ×20 magnification were collected for each membrane. Digital images were batch imported into ImageJ, thresholded and analysed with the Analyze particles function. Experiments were repeated three times. Where indicated, cell proliferation was assayed under identical conditions by a CellTiter 96 AQueous One Solution Cell Proliferation Assay or by direct counting of cells.

**Immunohistochemistry.** Immunohistochemical (IHC) detection of SNPH was performed on breast cancers and Cancer Universal Tissue Microarrays (CaU-TMAs) as described previously[44]. Briefly, 4 μm thick slides were incubated for 1 h with a rabbit polyclonal anti-SNPH antibody (Sigma#HPA049393 diluted 1:300). IHC was performed using an automatic stainer (Benchmark ULTRA, Ventana), with detection of antibody reactivity with a kit using peroxidase-diaminobenzidine as the chromogen (DAB UltraView, Ventana). All slides were counterstained with hematoxylin. As negative control, one slide was processed without the primary antibody. Since SNPH staining intensity was homogeneous among different samples types, IHC scores were attributed considering only the percentage of stained normal or tumour cells in each core (range 0–100%).

**Animal studies.** Studies were carried out in accordance with the recommendations in the Guide for the Care and Use of Laboratory Animals of the National Institutes of Health (NIH). Protocols were approved by the Institutional Animal Care and Use Committee (IACUC) of The Wistar Institute. Experiments were done without randomization, and without blinding. Sample size was determined by power analysis. All animals were included in the analysis. A liver metastasis model was performed essentially as described[8]. Surgical procedures were carried out in isoflurane-anaesthetised animals following aseptic technique inside a biosafety cabinet, and a slow release Buprenorphine formulation was administered for pain relief. PC3 cells at 80% confluency were suspended in PBS, and 50 μl containing $1 \times 10^6$ cells were injected intrasplenically in anaesthetised 6–8 weeks old male NOD SCID γ (NSG, NOD.Cg-*Prkdc*[scid] *Il2rg*[tm1Wjl]/SzJ) mice (Jackson Laboratory).

Spleens were removed 24 h after injection to minimize potentially confounding effects on metastasis due to variable growth of primary tumours. Animals (three per group) were euthanized 11 days after injection, and livers were resected, fixed in formalin and paraffin embedded. Serial liver sections 500 μm apart ($n = 15$ per each condition) were stained with hematoxylin and eosin and analysed histologically. Metastatic foci were quantified by morphometry and expressed as number of lesions and surface areas of tumour growth compared with total surface area, as described[8].

**Patients.** Breast cancer tissues from 324 female patients who underwent curative surgery at Fondazione IRCC Ca' Granda between 2004 and 2008 were retrieved from the archives of the Division of Pathology. The study was approved by an Institutional Review Board at Fondazione IRCCS Ca' Granda Ospedale Maggiore Policlinico Milan, Italy (approval no. 179/2013). Informed consent was obtained from all subjects. Representative cores of normal mammary glands ($n = 114$), carcinoma in situ lesions (CIS, $n = 79$), as well as of tumour bulk (core lesion) and invasive front (IF, $n = 299$) were arranged in tissue microarrays as previously described[44]. The patients' clinicopathological features according to the American Joint Committee on Cancer -AJCC- TNM system are summarized in Supplementary Table 2. Breast cancer metastatic burden to lymph nodes was calculated as the proportion of metastatic nonsentinel lymph nodes over the total number of analysed lymph nodes. The value of 10% was chosen as cutoff for high metastatic burden (Met LN > 10%).

Breast cancer molecular subtypes were determined by IHC assays of ER, PR, HER2 and Ki67 markers performed at diagnosis following the 2011 St Gallen guidelines[45,46]. Data were available for 321 patients (99%), who were therefore classified as luminal A ($n = 182$), luminal B-HER2 negative ($n = 95$), basal-like ($n = 40$), or HER2-enriched ($n = 4$) subtypes. For a subgroup of breast cancer patients ($n = 10$), DNA-free total RNA was available from normal, CIS, tumour bulk and invasive front lesions. These samples were investigated for presence of SNPH isoforms by PCR on retrotranscribed cDNA. A Cancer Universal Tissue Microarray (CaU-TMA) platform has been described[21,44].

**Statistical analyses.** Unless otherwise stated, all experiments were carried out in triplicates and repeated at least thrice ($n = 3$). For single cell analyses, the sample sizes are indicated in each figure and represent the cumulative n from a minimum of three independent experiments or time lapses. For descriptive data analysis, means, s.d. and medians were calculated, and distributions of data were examined to ascertain whether normal theory methods were appropriate. Student's t-test or Wilcoxon rank sum test was used for two-group comparative analyses. For multiple-group comparisons, ANOVA or Kruskal–Wallis test with post-hoc Bonferroni's procedure were applied. Variance similarity between groups was tested with Fisher (two groups) or Bartlett's (multiple groups) tests. All statistical analyses were performed using GraphPad software package (Prism 6.0) for Windows. A P value of < 0.05 was considered as statistically significant.

**Code availability.** The custom software for segmenting and tracking mitochondria can be downloaded from https://git-bioimage.coe.drexel.edu. The program codes are provided free and open source, under the BSD license. To run the software, MATLAB version 2015b or later is required.

**Data availability.** The data that support the findings of this study are available from the corresponding author upon request.

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

## Acknowledgements

The authors wish to thank David Schultz for advice with shRNA screening, Shashi Bala for cDNA library preparation for deep sequencing, and Frederick Keeney for assistance with time lapse microscopy. This work was supported by National Institutes of Health (NIH) grants P01 CA140043 (D.C.A., D.I.G. and L.R.L.), R01 CA78810 and CA190027 (D.C.A.), R01 CA089720 (L.R.L.), R01 NS076709 (A.R.C.), F32 CA177018 (M.C.C.), T32 CA009171 (K.G.B.), the Italian Minister of Health grant GR2011-02351626 (V.V.), the Office of the Assistant Secretary of Defense for Health Affairs through the Prostate Cancer Research Program under Award No. W81XWH-13-1-0193 (D.C.A.), and a Challenge Award from the Prostate Cancer Foundation (PCF) to M.C.C., D.I.G., L.R.L. and D.C.A. A.M. is supported by a fellowship from the Doctorate School of Molecular and Translational Medicine at the University of Milan. Support for Core Facilities utilized in this study was provided by Cancer Center Support Grant (CCSG) CA010815 to The Wistar Institute.

## Author contributions

M.C.C. and D.C.A. conceived the project; M.C.C. performed experiments of shRNA screening, mitochondrial trafficking, chemotaxis, focal adhesion dynamics, tumour cell invasion and the mouse model of metastasis; J.H.S. performed biochemical fractionation of mitochondria; A.A., J.E.H. and E.W. analysed time-lapse microscopy data of subcellular mitochondrial dynamics; K.G.B., A.M., S.F. and V.V. analysed primary patient samples; A.V.K. analysed bioinformatics data; M.C.C., S.B., D.I.G., L.R.L. A.R.C. and D.C.A. analysed data, and M.C.C., A.R.C. and D.C.A. wrote the paper.

## Additional information

**Competing financial interests:** The authors declare no competing financial interests.

