## [Peer Review File · Nature Communications]

Reviewers' comments:

Reviewer #1 (Remarks to the Author):

Starting from an shRNA screen, this manuscript by Caino and colleagues, identifies SNPH as a key molecule that regulates mitochondrial motility in cancer cells, this impacts on cancer cell migration, metastasis and ultimately on patient survival. Loss of SNPH expression promotes mitochondrial dynamics, motility and peri-nuclear mitochondrial accumulation, where they are proposed to power cellular movement and migration. Regulation of mitochondrial migration and dynamics its impact on cell migration is becoming an area of intense interest in cancer biology. In my opinion this manuscript offers new molecular insight into the process and the data largely support the authors' conclusions. I do have a number of points that should be addressed

1 - while I realise the shRNA screen ultimately forms a small part of this study I found it surprising that knockdown of SNPH could promote migration in the presence of gamitrinib given that this inhibitor blocks oxidative phosphorylation, given what we know about SNPH its difficult to see how it's knockdown could revert this effect - the authors should comment on this.

2 -SNPH clearly impacts on mitochondrial function in numerous ways. While perinuclear accumulation (generating ATP at a relevant site) may be important for driving metastasis, it is not directly tested and other possibilities exist - for example does reduction of SNPH increase mitochondrial number per cell and/or ATP levels per cell ? - these possibilities should be investigated

3 - The title should be amended, its clear from the authors own data that in many tissues/cell types this is not a reawakening on a neuronal network since it already pre-exists.

Reviewer #2 (Remarks to the Author):

In this manuscript, Caino et al. provide compelling evidence that mitochondrial dynamics, including mitochondrial trafficking, repositioning, fission and fusion, regulate the bioenergetics requirements of tumor cell invasion and metastasis. Through analysis of shRNA screening results, the authors identified SNPH, which is considered to moderate mitochondrial trafficking in neurons, as a non-neuronal tissue specific factor to suppress cancer cell invasion. Using multi-disciplinary cell biological, video graphic, the in vivo studies, and human clinical studies, SNPH was revealed to block cell motility and further tumor metastasis by regulation of reprogramming of mitochondrial dynamics.

While previous studies have reported mitochondrial metabolism as a tumor driver for supporting tumor cell motility, this study provided new mechanisms of SNPH-mediated mitochondrial dynamics to fuel chemotaxis and enhance metastatic competence. Moreover, the authors provided evidence from public databases and clinical samples that SNPH levels are decreased in different types of human tumors and low SNPH levels correlate with worse patient prognosis. Overall this study demonstrated a new mechanism by which tumor cell invasion is regulated by a SNPH-mediated pathway. While interesting and potentially important, there are some concerns that need to be addressed.

Major comments:

1. In Fig. 1d, 1e and Supplementary Fig. 1f, 1g, the authors compared depletion or overexpression of SNPH with control group in PC3 cells. They showed that SNPH significantly modulated tumor cell invasion by counting the number of invasive cells. However, the authors did not test whether SNPH significantly affected the total cell population in above conditions, which is necessary to

exclude the interference of cell proliferation.

2. Fig. 1e shows that knockdown of SNPH increased, while forced expression of SNPH decreased, cell invasion in PC3 cells. However, both control siRNA (siRNA Ctrl) and control vector (plasmid Ctrl) should not influence cell invasion. Thus, the authors need to clarify why the plasmid Ctrl group showed a significant increase in the number of invaded cells compared with the siRNA Ctrl (changed more than 2 folds).

3. The results showed that depletion of SNPH enhanced the rates of both fusion and fission events by increasing MFN1/2 protein expression and promoting Drp1 recruitment to mitochondria in tumor cells (Fig. 3a and Supplementary Fig. 5a-d). Could the authors provide additional experiments, or at least explanation, to account for the molecular mechanism by which SNPH increases MFN1/2 expression and promotes Drp1 recruitment to mitochondria?

4. In the experiments of Fig. 3c, forced expression of GD-Drp1 or GD-MFN2 significantly rescued the stimulating effect on tumor cell invasion in SNPH-depleted cells. However, the authors should explain why GD-MFN1 had no effect in this process, since MFN1, which was shown increased in SNPH-depleted cells, acts as a critical mediator of mitochondrial fusion.

5. To strengthen the connection between SNPH and KIF5B/Mico2 network (Fig. 3e-i), the authors should perform additional cell invasion experiment by silencing of KIF5B or Miro2 (or expression of their GD-mutant) in SNPH-depleted tumor cells.

6. The authors performed their most of mechanistic studies or the in vitro experiments by using some types of tumor cell lines, such as PC3 and LN229. However, the authors seem to jump from one cell line to another, but not consistently use the same cell lines to repeat at least the major experiments. For example: 1) In Fig. 1 and Supplementary Fig. 1, PC3 cells were used to perform tumor cell invasion assay, while FA complex dynamics by time lapse microscopy was only detected in LN229 cells. 2) To examine mitochondrial fission and fusion, all the function experiments were performed in LN229 cells (Fig. 3a and Supplementary Fig. 5a, 5b), while PC3 cells were only used to detect fission/fusion related protein levels by Western blot (Supplementary Fig. 5c-f).

As PC3 cell line is considered an appropriate model for cell migration experiments, the authors should carry out some key experiments, like measurement of FA complex dynamics and mitochondrial dynamics, in PC3 cells.

Minor comments:

1. The ordinate unit of Supplementary Fig. 1f and 1g should be consistent (invaded cells or invaded area).

2. Typo in line 94 of the main text: "Fig. 2h" should be "Fig. 1h";

Typo in Fig. 3g of the main text: "KIF5" should be "KIF5B";

Typo in line 437 of the supplementary text: "(e) and F" should be "(e, f)".

3. Supplementary Fig. 5i is not mentioned in the manuscript.

4. In Supplementary Fig. 6c, please explain what kind of condition "CM" refers to in the experiments.

5. The scale or magnification of images from tissue sections (e.g., Fig. 4b, 4f) need to be indicated.

RESPONSE TO REVIEWERS

Reviewer #1. We appreciate the thoughtful and constructive criticisms of this reviewer and we concur with the issues raised.

1) While I realise the shRNA screen ultimately forms a small part of this study I found it suprising that knockdown of SNPH could promote migration in the presence of gamitrinib given that this inhibitor blocks oxidative phosphorylation, given what we know about SNPH its difficult to see how it's knockdown could revert this effect - the authors should comment on this.

For the shRNA screen, we used a suboptimal concentration of Gamitrinib that provides partial inhibition of oxidative phosphorylation and is non-toxic for tumor cell types. This point has now been further clarified in the revised manuscript. In addition, we have uncovered here a novel role of SNPH in modulating mitochondrial shape dynamics (fusion and fission). This process generates new organelles, facilitates mitochondrial trafficking and quality control. As a whole, the opposing forces of fusion and fission allow for increased mitochondrial fitness in the face of energetic demands and metabolic stress (e.g. Gamitrinib treatment). Therefore, a potential mechanism by which SNPH depletion desensitizes tumor cells to Gamitrinib is the partial rescue of mitochondrial function provided by increased cycles of fusion/fission. This point has been further discussed in the revised manuscript as recommended by the reviewer.

2) SNPH clearly impacts on mitochondrial function in numerous ways. While perinuclear accumulation (generating ATP at a relevant site) may be important for driving metastasis, it is not directly tested and other possibilities exist - for example does reduction of SNPH increase mitochondrial number per cell and/or ATP levels per cell? - these possibilities should be investigated

We appreciate the point of the reviewer. In new experiments included in the revised manuscript, we now demonstrate that SNPH knockdown does not affect mitochondrial mass (*new Supplementary Fig.Ii*) and is actually associated with reduced ATP production in tumor cells (our unpublished observations). This suggests that increased invasion in SNPH-targeted cells is associated with increased mitochondrial dynamics (fusion, fission and transport) without changes in mitochondrial mass.

3) The title should be amended, its clear from the authors own data that in many tissues/cell types this is not a reawakening on a neuronal network since it already pre-exists.

The title of the revised manuscript has been changed, as requested by the reviewer.

Reviewer #2. We appreciate the thoughtful and constructive criticisms of this reviewer and we concur with the issues raised.

Major comments:

1) *In Fig. 1d, 1e and Supplementary Fig. 1f, 1g, the authors compared depletion or overexpression of SNPH with control group in PC3 cells. They showed that SNPH significantly modulated tumor cell invasion by counting the number of invasive cells. However, the authors did not test whether SNPH significantly affected the total cell population in above conditions, which is necessary to exclude the interference of cell proliferation.*

We have now analyzed the effect of SNPH depletion on tumor cell proliferation, as requested by the reviewer. Our results show that SNPH knockdown either has no effect (shRNA) or modestly reduced tumor cell proliferation (siRNA) during the course of the assay. These results are presented in a **new Supplementary Fig. 1h** in the revised manuscript, and rule out that the increased tumor cell invasion associated with SNPH knockdown is due to changes in cell number.

2) *Fig. 1e shows that knockdown of SNPH increased, while forced expression of SNPH decreased, cell invasion in PC3 cells. However, both control siRNA (siRNA Ctrl) and control vector (plasmid Ctrl) should not influence cell invasion. Thus, the authors need to clarify why the plasmid Ctrl group showed a significant increase in the number of invaded cells compared with the siRNA Ctrl (changed more than 2 folds).*

As requested by the reviewer, we have now repeated our tumor cell invasion assays using the same pCMV6-myc-Flag plasmid utilized for expression of SNPH (pCMV6-SNPH-myc-Flag plasmids). The results of these experiments are shown in a **new Fig. 1d,e** in the revised manuscript, and provide more accurate normalization of the basal level of Matrigel invasion of control transfectants. As the reviewer will appreciate, the transfection conditions between plasmid and siRNA expression differ significantly, potentially accounting for small differences in background levels of cell invasion.

3) *The results showed that depletion of SNPH enhanced the rates of both fusion and fission events by increasing MFN1/2 protein expression and promoting Drp1 recruitment to mitochondria in tumor cells (Fig. 3a and Supplementary Fig. 5a-d). Could the authors provide additional experiments, or at least explanation, to account for the molecular mechanism by which SNPH increases MFN1/2 expression and promotes Drp1 recruitment to mitochondria?*

Mitofusin levels have been shown to increase in response to oxidative stress. Consistent with this possibility, we now demonstrate in a **new Fig. 3d,e** in the revised manuscript that expression of antioxidant peroxiredoxin III (Prx-III) prevents the increase in MFN1 levels in SNPH-depleted cells. Further, we demonstrate in a **new Supplementary Fig. 5i** in the revised manuscript that this response does not involve transcriptional changes, as mRNA levels of MFN1 or MFN2 were unaffected in SNPH-depleted cells.

4) *In the experiments of Fig. 3c, forced expression of GD-Drp1 or GD-MFN2 significantly rescued the stimulating effect on tumor cell invasion in SNPH-depleted cells. However, the authors should explain why GD-MFN1 had no effect in this process, since MFN1, which was shown increased in SNPH-depleted cells, acts as a critical mediator of mitochondrial fusion.*

We speculate that MFN1-GD may not function as a bona fide dominant negative mutant in the SNPH pathway. To account for this possibility, we have now repeated the experiments of MFN targeting using validated siRNA sequences in reconstitution experiments with SNPH-depleted cells. The results of these experiments are presented in a **new Fig. 3c** and **new Supplementary Fig. 5g,h** in the revised manuscript and demonstrate that both MFN1 and MFN2 are required to support the increase in tumor cell invasion induced by SNPH depletion.

5) *To strengthen the connection between SNPH and KIF5B/Miro2 network (Fig. 3e-i), the authors should perform additional cell invasion experiment by silencing of KIF5B or Miro2 (or expression of their GD-mutant) in SNPH-depleted tumor cells.*

We concur with the point of the reviewer. Accordingly, we have now re-investigated the participation of Miro1/Miro2 and KIF5B in the SNPH pathway using reconstitution experiments in SNPH-depleted cells. These results are presented in a **new Fig. 3g** and **new Supplementary Fig. 6a-c** in the revised manuscript, and demonstrate that Miro1 and KIF5B are required to sustain the increased tumor cell invasion in SNPH-depleted cells. Conversely, Miro2 depletion does not attenuate tumor cell invasion upon SNPH loss. To further characterize this response, we also examined the contribution of Miro1, Miro2 and KIF5B on mitochondrial trafficking mediated by cellular stress induced by small molecule PI3K inhibitors, as described in our recent studies (PNAS 112:8638). The results of these experiments are presented in a **new Fig. 3i** and **new Supplementary Fig. 6e,f** in the revised manuscript, and demonstrate that all three molecules (Miro1, Miro2 and KIF5B) are required for mitochondrial trafficking and tumor cell invasion under stress conditions. These results suggest that Miro2 contributes to mitochondrial trafficking in tumors independently of SNPH, and this point is now discussed in the revised manuscript.

6) *The authors performed their most of mechanistic studies or the in vitro experiments by using some types of tumor cell lines, such as PC3 and LN229. However, the authors seem to jump from one cell line to another, but not consistently use the same cell lines to repeat at least the major experiments. As PC3 cell line is considered an appropriate model for cell migration experiments, the authors should carry out some key experiments, like measurement of FA complex dynamics and mitochondrial dynamics, in PC3 cells.*

We concur with the point of the reviewer. Accordingly, new experiments of mitochondrial fusion and fission as well as FA dynamics have now been repeated in the model of PC3 cells. These results are presented in **new Fig. 2d, Fig. 2f, Fig. 2h, Supplementary Fig. 2b-e and Supplementary Fig. 5c,d** in the revised manuscript and further independently validate the earlier findings with LN229 cells.

Minor comments:

1. *The ordinate unit of Supplementary Fig. 1f and 1g should be consistent (invaded cells or invaded area).*

These are, in fact, two different assays for cell invasion, which have now been better explained in the Experimental Procedures section of the revised manuscript. The two

independent assays have been carried out to further strengthen the conclusion that SNPH depletion results in increased tumor cell invasion.

2. *Typo in line 94 of the main text: "Fig. 2h" should be "Fig. 1h";*

Typo in Fig. 3g of the main text: "KIF5" should be "KIF5B";

Typo in line 437 of the supplementary text: "(e) and F" should be "(e, f)".

We apologize for the inaccuracies. The typos have now been corrected in the revised manuscript.

3. *Supplementary Fig. 5i is not mentioned in the manuscript.*

This has now been corrected in the revised manuscript.

4. *In Supplementary Fig. 6c, please explain what kind of condition "CM" refers to in the experiments.*

CM stands for conditioned medium utilized as a chemoattractant. This has now been explained in the revised manuscript to improve the clarity of the presentation.

5. *The scale or magnification of images from tissue sections (e.g., Fig. 4b, 4f) need to be indicated.*

We apologize for the omission. Scale bars has now been provided in the revised manuscript.

REVIEWERS' COMMENTS:

Reviewer #1 (Remarks to the Author):

The reviewers have comprehensively addressed all my points.

Reviewer #2 (Remarks to the Author):

In the revised version, the authors have provided comprehensive response to my comments of the original manuscript. The additional experiments clarified that both Miro1 and KIF5B are involved in SNPH-mediated tumor cell invasion (Fig. 3g and Fig. S6c). Also, function experiments of FA and mitochondrial dynamics have been repeated in PC3 cells which support the findings in LN229 cells. Overall the authors have sufficiently addressed my concerns.